

# Transcranial sonography findings related to depression in parkinsonian disorders: cross-sectional study in 126 patients

Angela E.P. Bouwmans[1], Wim E.J. Weber[2], Albert F.G. Leentjens[3] and Werner H. Mess[4]

[1] Department of Neurology, Reinier de Graaf Gasthuis, Delft, Netherlands
[2] Department of Neurology, Maastricht University Medical Centre, Maastricht, Netherlands
[3] Department of Psychiatry, Maastricht University Medical Centre, Maastricht, Netherlands
[4] Department of Clinical Neurophysiology, Maastricht University Medical Centre, Maastricht, Netherlands

## ABSTRACT

**Background.** Transcranial sonography (TCS) has emerged as a potential diagnostic tool for Parkinson's disease. Recent research has suggested that abnormal echogenicity of substantia nigra, raphe nuclei and third ventricle is associated with increased risk of depression among these patients. We sought to reproduce these findings in an ongoing larger study of patients with parkinsonian syndromes.

**Methods.** A total of 126 patients with parkinsonian symptoms underwent the Hamilton Depression Scale, and TCS of the substantia nigra (SN) ($n = 126$), the raphe nuclei (RN) ($n = 80$) and the third ventricle ($n = 57$). We then calculated the correlation between depression and hyper-echogenic SN, hypo-echogenic RN and a wider third ventricle.

**Results.** In patients with PD we found no significant difference of the SN between non-depressed and depressed patients (46% vs. 22%; $p = 0.18$). Non-depressed patients with other parkinsonisms more often had hyperechogenicity of the SN than depressed patients (51% vs. 0%; $p = 0.01$). We found no relation between depression and the echogenicity of the RN or the width of the third ventricle.

**Conclusions.** In patients with parkinsonian syndromes, we found no association between depression and hyper-echogenic SN, hypo-echogenic RN or a wider third ventricle, as determined by transcranial sonography.

Corresponding author
Wim E.J. Weber,
wim.weber@mumc.nl

## INTRODUCTION

Idiopathic Parkinson's disease (PD) is the second most common neurodegenerative disease with a worldwide prevalence of 41–1,903 per 100,000 (*Pringsheim et al., 2014*). Diagnosis, especially in the early stages, is difficult, as there is no definitive diagnostic test. Over the last 10 years transcranial sonography (TCS) of the substantia nigra (SN) has emerged as a promising tool in this regard. Numerous ultrasound studies have found that a significant percentage of patients with IPD have a typical enlarged area of echogenicity in the substantia nigra (SN+), which is thought to be associated with increased iron concentrations (*Vlaar et al., 2009*).
Although PD is mostly known for its motor symptoms, it has now become clear that non-motor symptoms, such as depression, often contribute to the burden of disease (*Reijnders et al., 2008*). Depression has a major impact on PD patients: depressed PD patients have worse motor function, more cognitive symptoms, and a lower quality of life (*Reijnders et al., 2008*; *Schrag, 2006*; *Schrag et al., 2010*). The pathogenesis of depression in PD is still unknown. Studies have suggested that the serotonergic raphe nuclei (RN) might be involved (*Becker et al., 2001*; *Chagas et al., 2013*; *Kostic & Filippi, 2011*; *Leentjens, 2004*; *Palhagen et al., 2008*).

Sonography researchers have thus investigated the RN in PD patients, and reported that its echogenicity was reduced in depressed PD patients compared to non-depressed PD patients and healthy control subjects (*Becker et al., 1997*; *Berg et al., 1999b*; *Cho, Baik & Lee, 2011*; *Stankovic et al., 2015*; *Zhang et al., 2016*). Additionally, SN hyperechogenicity and a wider third ventricle has also been reported to be associated with an increased risk of depression (*Krogias et al., 2011*; *Walter et al., 2007c*; *Walter, Skoloudik & Berg, 2010*). TCS could be clinically useful, as diagnosing depression in PD patients is difficult (*Bouwmans & Weber, 2012*; *Poewe & Luginger, 1999*; *Shulman et al., 2002*).

We recently finished a prospective cohort study on the diagnostic accuracy of TCS in early parkinsonian patients (*Bouwmans et al., 2013*), and we used this dataset to explore the association between depressive symptoms and echogenic features of the SN, RN, and third ventricle in PD patients.

## PATIENTS AND METHODS

### Design

This study was a cross-sectional study, nested within a prospective cohort study that aimed to test the diagnostic accuracy of TCS of the SN in patients who were referred to a neurologist by their general practitioner (GP) because of recent-onset parkinsonism of unclear origin (*Bouwmans et al., 2013*) The study protocol was published before patient inclusion started (*Vlaar et al., 2007*). The main finding of the cohort study was that the diagnostic accuracy of the echogenicity of the SN as a diagnostic test for early PD is not sufficient for routine clinical use. The Institutional Review Board (IRB) of Maastricht University Medical Centre approved the study (MEC 05–228, 4 April 2006), which was registered in the ClinicalTrials.gov database as NCT00368199. Both the TCS and the Hamilton Depression Scale assessments were pre-specified in the published protocol. However, this study of the association between the two was not prespecified, and is thus a posthoc analysis.

### Patients

We considered 283 consecutive patients with parkinsonism of unknown origin, who were referred to the neurology outpatient clinic of Maastricht University Medical Centre, Maastricht and the Orbis Medical Centre, Sittard, The Netherlands (Presently: Zuyderland Medical Centre). Patients who did not consent or those in whom a definite diagnosis could be made at the first visit ($n = 42$) were excluded from the study. Hence, 241 patients were included. Of these, another 69 were excluded: 24 patients who upon examination

did not present with clear parkinsonian symptoms or who presented with drug-induced parkinsonism, as well as 45 patients (18.7%) who did not have a sufficient bone window for an adequate TCS examination (See flowchart).

## Measures

After signing informed consent, all subjects underwent a structured interview and a neurological examination (*Bouwmans et al., 2013*; *Vlaar et al., 2007*). All tests were performed by a physician (Senior resident in neurology) not treating the patient and blinded for information in clinical records. Depressive symptoms were measured with the observer-rated 17 item Hamilton Depression Rating Scale (HAMD). This scale has a good reliability and validity, both in PD patients as well as in the general population. (*Leentjens et al., 2000*; *Schrag et al., 2007*). The score range is 0–52, with scores of 11 and higher suggesting clinically relevant depressive symptoms. Motor symptoms were measured with the Unified Parkinson's Disease Rating Scale (UPDRS-III) (*Movement Disorder Society Task Force on Rating Scales for Parkinson's Disease, 2003*).

Within two weeks of inclusion all patients underwent a TCS at the department of Clinical Neurophysiology of one of the two hospitals. In the Maastricht University Medical Center, visualization of the RN was included in the TCS protocol from the start of the study. One year later, measurement of the third ventricle was included as well. In the Orbis Medical Center, Sittard, only the SN was visualized.

TCS was performed using a SONOS 5500 system (Philips, Eindhoven, The Netherlands). The examination took place in a darkened room with the patient already lying on the examination table before the investigator entered the room, in order to minimize the risk of identification of a patient's clinical signs. Patient and investigator were instructed not to discuss symptoms or diagnoses.

TCS was performed bilaterally through the pre-auricular bone window with a 2–4 MHz phased array transducer. The quality of the bone window was scored as good, moderate or inferior.

Two different methods were applied for the evaluation of the echointensity of the SN. Firstly, the presence or absence of an obviously visible bilateral hyperechogenic SN was scored (qualitative method). The SN were scored as hyperechointens, not hyperechointens or inconclusive (=no typical configuration of hyperechointensity or low quality of the temporal bone window). Secondly, the area of an eventually hyperechogenic SN was measured quantitatively (quantitative method). Both the right and left SN were measured from both sides, i.e., both temporal bone windows. After encircling, the area was automatically calculated. A hyperechogenic area of at least 0.2 cm2 was classified as characteristic for PD. This was only performed when the hyperechogenicity was located within the anatomical distribution of the SN, meaning that it showed a typical oblique stripe-shaped configuration. Both the right and left SN were measured from both sides.

The RN were identified if they met the criteria of an anatomic structure equally echo-intense to the red nucleus and localized in the transverse plane of the midbrain with a length extending from anterior to posterior, not interrupted. Echogenicity of the RN was rated using a visual scoring system resulting in a semi-quantitative assessment. We scored the

RN as hypo-echogenic (RN−) when this structure had a reduced echogenicity compared to the surrounding brain structures or when the anatomic structure was interrupted. We scored the RN as hyperechogenic (RN+) on the TCS when it showed as an uninterrupted relatively echo-intense structure. The patient was scanned from both sides because of the bone window variability in quality of visualization of the RN from right to left. We used the best possible result, so if the RN was absent on one side, but visible at the other side, it was scored as hyperechogenic.

The transverse diameter of the third ventricle was measured from both sides on a standardized diencephalic examination plane.

Two years after inclusion, patients were re-examined by two movement disorder neurologists to obtain a final clinical diagnosis, using the official diagnostic criteria for the several parkinsonian disorders (*Gilman et al., 2008*; *Hughes et al., 1992*; *Litvan et al., 1996*; *Litvan et al., 1999*; *McKeith et al., 2005*), which served as a gold standard for our study. These investigators were blinded for all test results, and none of the neurologists had seen the patient before. They were asked to interview and examine the patient, as they would normally do during a routine neurologic consultation. The neurologists filled in the same standard form as had been done by the including investigator during the first visit of the patient, which included, among others, the Unified Parkinson's Disease Rating Scale (UPDRS)-III score. Afterwards the neurologists received these scores of the patient at the first visit, so that they could evaluate whether the patient had had any progression on that scale. Each neurologist was then asked to reach a final clinical diagnosis of the parkinsonian syndrome. One investigator compared these scores and when there was no agreement, the two neurologists were asked to discuss these patients using their notes, in an effort to reach agreement on the final diagnosis (*Bouwmans et al., 2013*).

## STATISTICS

SPSS 21.0 for Windows was used for the statistical analysis. Comparing categorical variables was done by chi-square test. The two-sample $t$-test was used for comparing continuous variables. Before performing a post-hoc test, we used the homogeneity of variances to decide which post-hoc test was suitable. When showing a good homogeneity, we chose the Bonferroni or Tamhane T2 test for further analyses, otherwise. $P$ values of <0.05 were considered significant.

## RESULTS

### Patient characteristics

We allocated patients who were eventually diagnosed with essential tremor (ET) in the group of parkinsonism, because of the pathophysiologic resemblance with PD (*Fekete & Jankovic, 2011*; *Louis & Ottman, 2013*; *Shahed & Jankovic, 2007*; *Tan et al., 2008*). We had an insufficient bone window in 18.7% of our patients. This is in line with earlier studies that report an insufficient bone window in 10–20% of participants, or even up to 59% of women over 60 years (*Okawa et al., 2007*; *Walter et al., 2007a*).

**Table 1  Patient characteristics divided by final diagnoses.**

|  | PD ($n = 72$) | Other parkinsonisms ($n = 54$) | P value |
|---|---|---|---|
| Age, years (SD, CI) | 68.6 (9.2, 66.43–70.76) | 72.2 (9.3, 69.69–74.79) | *0.03* |
| Disease duration, months (SD, CI) | 30.1 (47.1, 19.01–41.16) | 41.7 (41.4, 30.42–53.02) | 0.15 |
| UPDRS total score, mean (SD, CI) | 24.5 (10.6, 21.99–27.00) | 10.6 (15.6, 24.94–33.60) | 0.06 |
| UPDRS motor score, mean (SD, CI) | 13.2 (5.7, 11.82–14.55) | 15.0 (7.8, 12.85–17.15) | 0.16 |
| HAMD, mean (SD, CI) | 4.6 (5.5, 3.33–5.92) | 5.8 (5.5, 4.28–7.28) | 0.25 |
| DP+ % | 12.5 | 13.0 | 0.94 |
| SN+ % | 43.06 | 44.4 | 0.88 |
| RN−% | 21.7 | 17.7 | 0.65 |
| Third ventricle width, mm (SD, CI) | 5.4 (2.10, 4.62–6.17) | 5.3 (2.38, 4.36–6.28) | 0.90 |

Notes.

PD, Parkinson's disease; UPDRS, Unified Parkinson's Disease Rating Scale; DP+, depression present/having a total score of the HAMD of 11 or more; SN+, hyperechogenic substantia nigra; RN−, hypo-echogenic raphe nuclei; SD, standard deviation; CI, confidence interval.

Eventually, we were able to obtain interpretable TCS images of the SN in 126 patients with HAMD rating scale scores. We did TCS of the RN in only one of the two hospitals, so in the end we had 80 patients with RN TCS images. Only later on in the study did we start measuring the width of the third ventricle (An amendment to the study protocol was made), so this echo feature was available for only 57 of the 126 patients.

At follow-up, 72 (57%) patients were clinically diagnosed with PD. 19 (15%) patients had atypical parkinsonian syndromes (APS), such as multiple system atrophy (MSA), progressive supranuclear palsy (PSP), Lewy body dementia (LBD) and corticobasal degeneration (CBD). Nineteen (15%) patients had vascular parkinsonism (VP) and 16 (13%) were diagnosed with ET (See Table 1). The subgroups differed significantly on a number of variables, with PD patients being younger and having higher (worse) UPDRS scores (Table 1). The average HAMD scores did not differ between the groups. Sixteen (13%) of the patients had a HAMD $\geq$ 11, indicating clinically relevant depressive symptoms. The percentage of patients with a hyperechogenic SN and the percentage with hypo-echogenic RN, did not differ between the group with PD and the one with other parkinsonisms. We compared actual HAMD scores between two groups (hyperechogenicity and hypo-echogenicity) with a $t$-test, but were unable to find a significant association (SN+ mean 4.22, SN− 5.82, $t = 1.70$, $p = 0.92$). The width of the third ventricle was also not significantly different between the two groups.

**Depression and echo features in PD and other parkinsonisms**
Nine (13%) of the 72 patients with PD had a HAMD > 11 versus 7 of the 4 patients with other parkinsonisms. There were no differences in the three TCS features between depressed and non-depressed PD patients. In patients with other parkinsonisms we found a significantly higher frequency of hyperechogenic SN in non-depressed patients (See Table 2). There were no significant differences in echo features of RN and the third ventricle between depressed and non-depressed patients with other parkinsonisms (See Figs. 1–3).

**Table 2  Patient characteristics divided by presence or absence of depression.**

| | PD and absence depression (*n* = 63) | PD and presence depression (*n* = 9) | *P* value | Other parkinsonisms and absence depression (*n* = 47) | Other parkinsonisms and presence depression (*n* = 7) | *P* value |
|---|---|---|---|---|---|---|
| Mean age, years (SD, CI) | 69.8 (8.8, 67.04–71.50) | 63.9 (10.9, 55.52–72.26) | 0.10 | 71.9 (9.4, 69.18–74.70) | 74.3 (9.4, 65.61–82.96) | 0.54 |
| Mean duration complaints, months (SD, CI) | 32.0 (49.8, 19.48–44.59) | 16.4 (14.6, 5.25–27.64) | 0.36 | 44.7 (42.7, 32.21–57.28) | 21.4 (24.8, −1.46–44.32) | 0.17 |
| UPDRS total score, mean (SD, CI) | 24.2 (10.3, 21.56–26.80) | 26.7 (12.8, 16.85–36.49) | 0.82 | 27.5 (14.3, 23.26–31.74) | 42.8 (19.7, 22.19–63.48) | *0.02* |
| UPDRS motor score, mean(SD, CI) | 13.3 (5.4, 11.86–14.63) | 12.8 (8.0, 6.61–18.95) | 0.82 | 14.3 (7.5, 12.06–16.45) | 20.8 (8.4, 12.02–29.65) | *0.05* |
| HAMD, mean (SD, CI) | 3.0 (3.2, 2.18–3.79) | 16.1 (4.4, 12.75–19.47) | *0.00* | 4.1 (2.7, 3.31–4.90) | 17.0 (6.5, 11.01–22.99) | *0.00* |
| SN+ % (CI) | 46.0 (7.1, 42.9) | 22.2 | 0.18 | 51.1 | 0 | *0.01* |
| RN−% (CI) | 21.1 (−28.7, 36.6) | 25.0 | 0.81 | 20.0 | 0 | 0.32 |
| Third ventricle width, mm (SD, CI) | 5.4 (1.9, 4.57–6.16) | 5.5 (3.0, 2.43–8.63) | 0.84 | 5.3 (2.5, 4.22–6.45) | 5.3 (1.7, 2.60–7.90) | 0.95 |

**Notes.**
SN+, hyperechogenic substantia nigra; RN−, hypo-echogenic raphe nuclei; SD, standard deviation; CI, confidence interval.

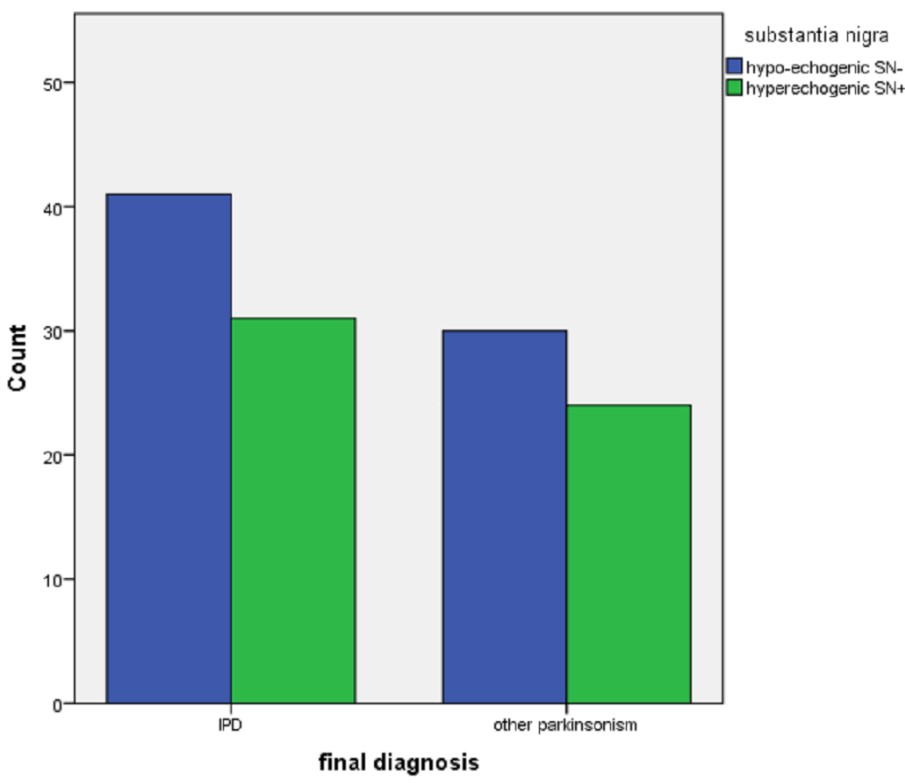

**Figure 1** Echogenicity of the substantia nigra in patients referred for parkinsonism, by depression ($N = 16$ with HamD > 10, $n = 110$ with HamD < 11) and by final diagnosis (IPD, $n = 72$; other, $n = 54$).

## DISCUSSION

In this cross-sectional study of 126 early stage parkinsonian patients we did not find any relation between the presence of depressive symptoms and the echogenicity of the SN, RN nor the width of the third ventricle. We found a higher frequency of hyperechogenicity of the SN in the non-depressed patients with other parkinsonisms, but the significance of this remains unclear as it is the result of a posthoc subgroup analysis.

The major limitation of our study is that it is a secondary analysis of a study that was not powered to detect differences in echogenicity of the SN and RN, or width of the third ventricle resulting from the severity of depressive symptoms. This resulted in a relatively low proportion of patients with clinically relevant depressive symptoms and subgroup analyses were done with a limited number of depressed patients. Because of this, our results must be seen as exploratory and interpreted with caution. In particular, the lack of a significant difference in the proportion of hyperechogenicity of the SN between depressed and non-depressed PD patients may be due to a lack of power. Another limitation is the lack of a formal psychiatric assessment to support a diagnosis of depression based on diagnostic criteria. However, a HAMD score $\geq 11$ is considered a good indicator of clinically relevant depressive symptoms and has been used to screen for depression in several studies (*Leentjens et al., 2000*; *Reijnders, Lousberg & Leentjens, 2010*; *Schrag, 2011*; *Schrag et al., 2007*). Finally, a limitation of all these studies is the lack of a gold standard clinical

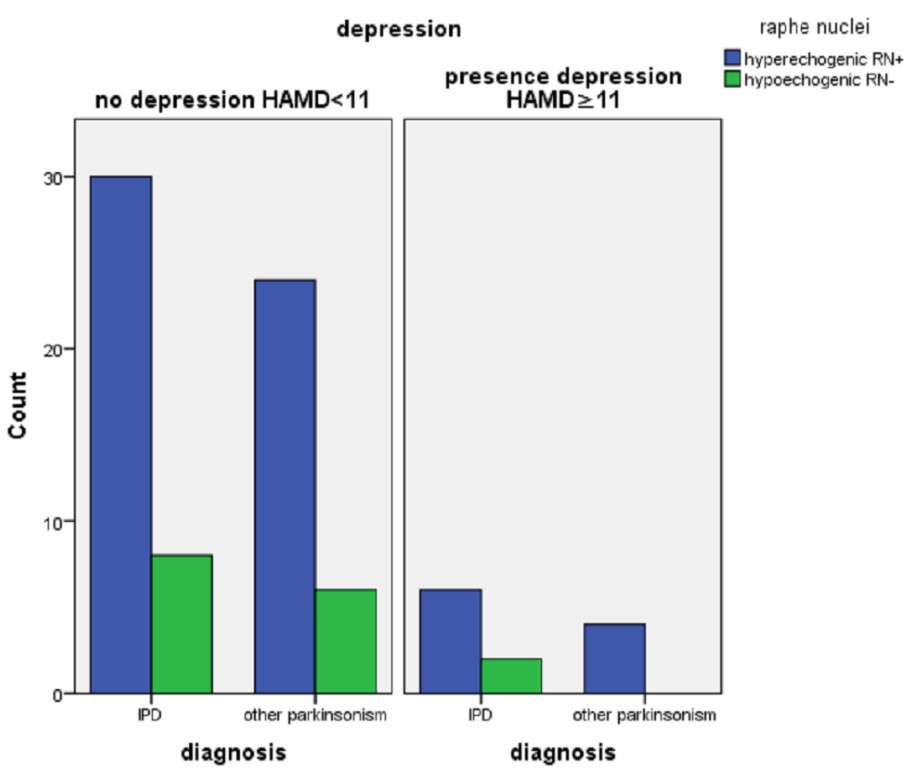

**Figure 2** Echogenicity of the raphe nuclei in patients referred for parkinsonism by depression ($n = 12$ with HamD > 10, $n = 68$ with HamD < 11), and by final diagnosis (IPD, $n = 46$; other ($n = 34$).

diagnosis. The accepted gold standard is postmortem neuropathological examination, which is shown to differ substantially even from movement disorder specialists' diagnoses (*Joutsa et al., 2014*). However, for practical reasons, clinical examination after several years is a reasonable surrogate. Another limitation is the two years follow-up. While longer follow-up is preferable, two years is enough to assess the three most important temporal diagnostic criteria for Parkinson's disease (*Hughes et al., 1992*).

As the RN are thought to play a role in the pathogenesis of depression, investigators have used TCS to visualize these structures in depressed patients. They found that depressed (*Becker et al., 1995a*; *Becker et al., 2001*; *Becker et al., 1994*; *Walter et al., 2007c*; *Walter et al., 2007d*) and bipolar patients (*Krogias et al., 2011*) have a reduced echogenicity of the RN compared to healthy controls. Others reported that PD patients with depression have lower RN echogenicity compared to non-depressed PD patients and healthy control subjects (*Becker et al., 1997*; *Berg et al., 1999b*; *Cho, Baik & Lee, 2011*; *Stankovic et al., 2015*; *Walter et al., 2007b*; *Zhang et al., 2016*) suggesting that hypo-echogenicity of the RN may be a sign of (preclinical) dysfunction of the limbic system. However, in our study we could not confirm these findings.

There are essential differences between the above five studies that did find an association and ours that did not. First, the definition of depression was different. Some of the previous studies used DSM-III and DSM-IV criteria to diagnose depressive disorder (not specified if this was done by a board-certified psychiatrist), and others used the Hamilton Depression

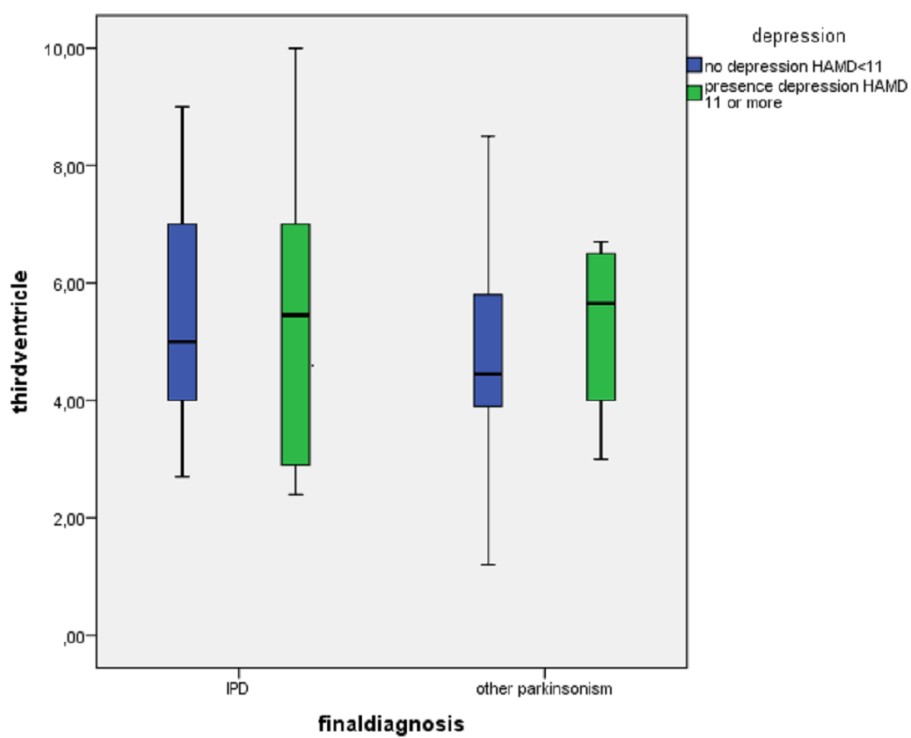

**Figure 3** Division of width of third ventricle between patients with ($n = 10$) and without a depression ($n = 47$) divided by diagnosis IPD ($n = 31$) and other parkinsonisms ($n = 26$).

and Montgomery–Asberg Depression Rating Scale, with varying cut-offs. Secondly, patient selection was different. We did a prospective study in which consecutive patients were enrolled. The other studies, with the exception of Stankovic's, did not describe how patients were recruited, and it is not clear whether this might have biased results (*Colditz, 2010*). Another difference is the disease duration of the included patients: with the exception of Cho's study, all studies included patients with a longer disease duration, up to 17 yrs. even (*Berg et al., 1999a*). This is possibly also reflected in the high percentages of patients with clinically relevant depressive symptoms in these studies: 43–47%, while this prevalence is generally assumed to be around 35% (*Reijnders et al., 2008*). This approach is useful in pilot experiments, but when one wants to assess diagnostic accuracy of a technique it is preferable to include patients who have not yet been definitely diagnosed (*Bachmann et al., 2009*). The inclusion of early, not yet diagnosed patients in our study may explain the low percentage (13%) of patients with clinically relevant depressive symptoms.

Some studies have suggested a relation between the width of the third ventricle and the presence of depression (*Krogias et al., 2011*), as enlargement of the third ventricle may be a reflection of the atrophy of the surrounding structures (*Hendrie & Pickles, 2010*). We were also not able to confirm these findings.

In a meta-analysis we did on TCS in parkinsonian syndromes we found that in 7 retrospective studies a decreased echo-intensity of the RN was found more often in depressed (46%) than in non-depressed IPD patients (16%). Our present study does not

accord with that observation. We hypothesize that one of the main reasons for this is publication bias, where negative studies on TCS tend not to be published. We did not formally test that in our meta-analysis, but another example is our recent negative study on the accuracy of TCS to diagnose IPD. Many studies (*Becker et al., 1995b*; *Berg, Siefker & Becker, 2001*; *Gaenslen et al., 2008*; *Huang et al., 2007*; *Kim et al., 2007*; *Mehnert et al., 2010*; *Ressner et al., 2007*; *Spiegel et al., 2006*; *Walter et al., 2002*) found a striking association between a hyperechogenic SN and the diagnosis of PD, but in a carefully designed and executed diagnostic accuracy study we could not confirm these results (*Bouwmans et al., 2013*).

In conclusion, we did not find any relation in early stage parkinsonian patients between the presence of depressive symptoms and the echogenicity of the SN, RN nor the width of the third ventricle. At present, this technique has limited diagnostic value to diagnose or predict depression in parkinsonian patients.

### Funding
This work was funded by the "Stichting Internationaal Parkinson Fonds," The Netherlands. The funders had no role in study design, data collection and analysis, decision to publish, or preparation of the manuscript.

### Competing Interests
The authors declare there are no competing interests.

### Author Contributions
- Angela E.P. Bouwmans conceived and designed the experiments, performed the experiments, analyzed the data, contributed reagents/materials/analysis tools, wrote the paper, prepared figures and/or tables, reviewed drafts of the paper.
- Wim E.J. Weber, Albert F.G. Leentjens and Werner H. Mess conceived and designed the experiments, performed the experiments, analyzed the data, contributed reagents/materials/analysis tools, wrote the paper, reviewed drafts of the paper.

### Human Ethics
The following information was supplied relating to ethical approvals (i.e., approving body and any reference numbers):

The Institutional Review Board (IRB) of the Maastricht University Medical Centre, MEC 05–228, 4 April 2006.

### Data Availability
The raw data has been supplied as Data S1.

### Supplemental Information
Supplemental information for this article can be found online at http://dx.doi.org/10.7717/peerj.2037#supplemental-information.

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
