# Peer review of "Transcranial sonography findings related to depression in parkinsonian disorders: cross-sectional study in 126 patients"

_PeerJ, doi:10.7717/peerj.2037_

## Round 0.1 · original submission · Major Revisions

The reviewers' recommendations ranged from "reject" to "minor revisions." I believe you report valuable data, especially given the state of the literature that you summarize in the introduction. Your prospective, blind study design and expert movement disorders assessments strengthen the importance of the work. You also published the study protocol in advance. However, the report will be much more convincing with some additional information and clarifications.

Most importantly, asserting a negative result places the onus on the authors of defending the power of the study for the specific result. Please provide confidence intervals for the main results.

You laudably include in the discussion the secondary nature of the depression analysis. Please clarify whether any of the depression analyses were specified in advance (e.g. included in the published protocol), and if so provide the specifics (e.g., association with depression in the whole group? in the final-iPD group? etc.) Most important in this context, why did you dichotomize the HamD? Did you pre-specify that approach? Did the previous reports use categorical variables? Analyzing the key imaging results with depression score as a continuous variable would be of great interest, but of course should be flagged as a secondary analysis if that is what it is. This seems especially relevant given the issues that arise from the small number of subjects in the HamD>10 group, as noted by the reviewers, but also because the HamD is much more defensible as a (roughly continuous) measure of symptom severity than as a proxy for (dichotomous) clinical diagnosis.

Add to the limitations section the recognition that the final diagnosis was not based on autopsy (as you acknowledged in your 2013 paper), and acknowledge the consequent limitations (e.g. published reports of antemortem clinical diagnosis of iPD at expert centers show something like 70-80% accuracy vs postmortem pathological diagnosis).

"Sixteen (13%) of the patients had a HAMD ≥ 11" -- That is a lower prevalence than almost all other studies. Any idea why? Similarly, from clinical experience and my possibly faulty recollection of the literature, I believe that the HamD scores in the non-depressed group (3-4 points, Table 2) are lower than is usually observed in non-depressed PD patients. At a minimum, please report who scored the HamD (i.e., what were the qualifications or training of the rater, and if available provide any evidence of accuracy or inter-rater reliability).

Line 240: Please comment on how the other 5 studies defined depression.

Please also address the methods questions from the reviewers. I'm not worried about the VPD and ET comment, but you should reply to the reviewer and consider mentioning it in discussion. Please reply to the reviewer about "whether affective disorders are the same between the involved neurological entitites," but in the context of this study, this question need not be addressed in the manuscript text.

Additional comments:
* lines 129: "the SN area was encircled" -- do you mean "the hyperechogenic area in the SN was encircled"? Similar question for the preceding sentence, line 128.
* lines 134-135: "The RN were identified if they met the criteria of an anatomic structure equally echo-intense to the red nucleus" -- ??
* Table 1: add SD to data in last row
* Bouwmans 2013 reference: The issue and article number are missing, and the DOI is incomplete.
* Figure 1 title: Suggest new title: "Echogenicity of the substantia nigra in patients referred for parkinsonism, by depression (N=16 with HamD>10, n=110 with HamD<11) and by final diagnosis (IPD, n=72; other, n=54). And ideally sort first by diagnosis and second by HamD scores, rather than the other way around as presented here, to facilitate the viewer's seeing the comparison of interest (i.e., depression).
* Figure 2: same comments.

Reviewer 1 ·

Basic reporting

No Comments

Experimental design

No Comments

Validity of the findings

No Comments

Additional comments

This paper is well designed, investigations are sound and the figure presenting the overall finding is convincing. The authors wanted to compare TCS findings in parkinsonian syndrome patients with depression and non-depression. They showed no differences between two groups, and this data is unlike of previous reports.
I have some comments of this study.
1. Authors should to define of hyperechogenicity of SN with more detailed. They didn't mention about cut off score of hyperechogenicity of SN in their method.
2. Authors divided into PD and atypical PD group. However, VPD and ET are different disease entity unlike other atypical PD (PSP, MSA, CBD) group. So patients with VPD and ET should be excluded or compared independently. If authors do that, atypical Parkinsonian syndrome group is too small ( 19 persons for MSA, PSP, CBD)
3. Authors talked the why that this results are different to previous reports is about duration of disease. However, I know that the echogenicity of SN is not related to disease duration.
4. the number of patients with depression is too small to compare between two groups by statistical analysis. (PD with depression: 9, and other PD with depression : 7)
4. There are some minor mistakes in manuscript. ( the title overlap : 3-4, omit reference ; 261)

Reviewer 2 ·

Basic reporting

The findings are somewhat different from the bulk of published data (i.e. SN hyperechogenicity is present in 43% and 44% of PD patients and patients with parkinsonism, respectively). The prevalence of hyperechogenic SN among PD patients is >80% in the majority of reports and significantly less (if any) in atypical parkinsonian syndromes (suggested to be one of the criteria for differentiation. Therefore, authors should discuss in more details these differences. Considering relatively small numbers among atypical parkinsonism, they also should explain if 2 years follow-up is long enough for diagnosis discrimination (i.e. PSP-P, MSA-P, etc.)

Experimental design

The authors mentioned basic limitations of the study.
However, they did not control for comorbid anxiety found to be important variable in determination of the RN status (i.e. a recently published study found an association of reduced midbrain raphe echogenicity with panic disorder (Silhan et al., Psychiatry Res. 2015 Oct 30;234(1):137-43), in partial continuation of the findings of Stankovic et al.).
The authors should describe how they have obtained their cut-of values for the SN hyperechogenicity

Validity of the findings

The findings are important since they arise controversies that may be useful for realistic positioning of TCS.
One possibility the authors should discuss is whether affective disorders are the same between the involved neurological entitites, particularly PD and atypical parkinsonism. They have to give us explanation for lumping them together.

Additional comments

No general comments.

·

Basic reporting

The article meets all requirements.

Experimental design

The study meets all requirements.

Validity of the findings

The articles succeeds all requirements too.

Additional comments

I would have only two notes:

(1) it would be advisable to reformulate the title of this article - to insert '...are not related', thus emphasizing the negative results of TCS as at present it seems that the TCS findings are related to depression.

(2) all pictures (graphs/charts) need to be corrected by inserting measurement units into x and y axes, correcting the spelling, also by supplying a better quality images where it is possible.

---

## Round 0.2 · Minor Revisions

Overall the revised submission is fine, but some of the previously requested changes were not addressed ideally, and doing so could still improve the paper substantially. Specifically:

1. Add confidence intervals: Thanks for the ones you added, but is it not possible to estimate confidence intervals for the key results in Table 2 ratios (SN+ and RN+)? I.e. you have a p value of 0.18 for [SN+ in PD,HamD<11 46.0% = 29 of 63], vs [SN+ in PD,HamD>10 22.2% = 2 of 9], but it would be nice to see a C.I. for the difference in proportions.
OK, I'm not the expert here but I think this is what you want: http://www.statisticslectures.com/topics/ciproportions/
If I'm doing it right, the 95% C.I. for the difference in SN+ rate is (7.1%, 42.9%) and for the difference in RNhypo rate is (-28.7%, 36.6%).

2. Page 11: Please add this brief summary of the t test results: "... unable to find a significant difference (SN+ mean x.xx, SN- y.yy, t=a.aa, p=0.bb)."

3. "But this [autopsy] is hardly feasible anymore in modern times, as relatives are reluctant to permit this. So, the methodologically highest achievable gold standard is clinical examination after several years."

There are two problems with this text.

First, in my opinion that view is way too pessimistic. I ask my patients about autopsy in advance and many say yes. A family helped make sure we got an autopsy just last week after the patient died. And autopsy is still the gold standard for a diagnostic study in PD. But I understand that the autopsy rate will be much lower than 100% under the best conditions, and besides hopefully it will be many years away from the diagnosis. Please modify this statement. Perhaps, if you agree, you could instead say something like "However, for practical reasons, clinical examination after several years is a reasonable surrogate."
Second problem: see #9, below.

4. "These qualities, if present, would be the most important risk factors for depression in PD (See: Leentjens et al. Neurology 2013;81:1036–1043), which are absent in de novo patients."
I disagree. Please consider modifying this statement. Leentjens et al 2013 in fact says: "Nonspecific risk factors had a 3-times-higher influence in the model than PD-specific risk factors. ... Accordingly, research on depression in PD should focus not only on factors associated with or specific for PD, but should also examine a wider scope of factors including general risk factors for depression, not specific for PD." Besides, depression occurs in higher rates in people with PD even before their diagnosis with PD.

5. "the prevalence [of depression in PD] is generally assumed to be 7-19% (Reijnders et al. 2008)."
Caution: you are mixing concepts here, leading to an erroneous conclusion. Specifically, in this manuscript you are not talking about major depression as a syndrome, you're talking about the prevalence of BDI scores higher than 10, and that prevalence is usually closer to 45%. Even the results of Reijnders et al. 2008 place this prevalence at between 35 and 52%: "Clinically significant depressive symptoms, irrespective of the presence of a DSM defined depressive disorder, were present in 35%," whereas "the weighted prevalence of major depressive disorder was 17% of PD patients, that of minor depression 22% and dysthymia 13%," for a total of 52%.
By contrast, the 19% you cite comes from Reijnders et al's discussion of _syndromal_ major depression. They state, "In studies using a (semi) structured interview to establish DSM criteria, the reported prevalence of major depressive disorder was 19%." This is consistent with my own reading of the evidence, which is that the cross-sectional prevalence of major depression in PD is 20-25%, with an additional 20-25% having other clinically relevant depressive symptoms (dysthymia, "minor depression," and subsyndromal depression).
Bottom line: I think you still need to recognize that the prevalence of depressive symptoms is much lower in your study than in most other studies. I don't know that you need to speculate in the ms. as to the cause.

6. Bouwmans 2013: The article number (e002613) is still missing in the bibliography. That's not surprising, as MEDLINE missed it, too, though PubMed Central has the complete reference. You may want to ask the NLM to fix their records.
The journal shows: BMJ Open 2013;3:e002613 doi:10.1136/bmjopen-2013-002613

7. Fig. 1: The old figure 1 showed 8 bars, reading, from left to right:
HamD<11,IPD,SN-; HamD<11,IPD,SN+; HamD<11,other,SN-; HamD<11,other,SN+;
HamD>10,IPD,SN-; HamD>10,IPD,SN+; HamD>10,other,SN-; HamD>10,other,SN+
In the resubmission I only see half of the original figure (the first 4 bars). What I was trying to ask for was a new figure with 8 bars, but in a different order, from left to right reading:
HamD<11,IPD,SN-; HamD<11,IPD,SN+; HamD>10,IPD,SN-; HamD>10,IPD,SN+;
HamD<11,other,SN-; HamD<11,other,SN+; HamD>10,other,SN-; HamD>10,other,SN+
This request is not mandatory, but I think it would make Fig. 1 and Fig. 2 much clearer.

8. Fig. 2: This is unchanged from the old figure 2.

9. Rev. 2 said: "Considering relatively small numbers among atypical parkinsonism, they [the authors] also should explain if 2 years’ follow-up is long enough for diagnosis discrimination (i.e. PSP-P, MSA-P, etc.)."
Please comment on this in Discussion (whether 2 years is long enough for a clinical "gold standard"). In other words, don't just say that clinical follow-up is the best we can do, but specifically discuss whether 2 years (as opposed to 3, 5 or 10 years) is long enough to be meaningful.

·

Basic reporting

No comments

Experimental design

No comments

Validity of the findings

No comments

Additional comments

I would still suggest to change the title of manuscript to emphasize that findings were not related.

Reviewer 4 ·

Basic reporting

.

Experimental design

.

Validity of the findings

.

Additional comments

The author have responded nicely to all my comments.

---

## Round 0.3 · accepted · Accept

Thank you for your attention to the points raised previously.

One note, at PeerJ you have the choice of making public the reviews and your rebuttals. Please consider doing so, as several interesting points were raised that the more interested reader may benefit from.